# The Role of miRNAs in Neuropathic Pain

**DOI:** 10.3390/biomedicines11030775

**Published:** 2023-03-03

**Authors:** Martina Morchio, Emanuele Sher, David A. Collier, Daniel W. Lambert, Fiona M. Boissonade

**Affiliations:** 1School of Clinical Dentistry, University of Sheffield, Sheffield S10 2TA, UK; 2The Neuroscience Institute, University of Sheffield, Sheffield S10 2TN, UK; 3UK Neuroscience Hub, Eli Lilly and Company, Bracknell RG12 1PU, UK

**Keywords:** microRNA, chronic pain, neuropathic pain

## Abstract

Neuropathic pain is a debilitating condition affecting around 8% of the adult population in the UK. The pathophysiology is complex and involves a wide range of processes, including alteration of neuronal excitability and synaptic transmission, dysregulated intracellular signalling and activation of pro-inflammatory immune and glial cells. In the past 15 years, multiple miRNAs–small non-coding RNA–have emerged as regulators of neuropathic pain development. They act by binding to target mRNAs and preventing the translation into proteins. Due to their short sequence (around 22 nucleotides in length), they can have hundreds of targets and regulate several pathways. Several studies on animal models have highlighted numerous miRNAs that play a role in neuropathic pain development at various stages of the nociceptive pathways, including neuronal excitability, synaptic transmission, intracellular signalling and communication with non-neuronal cells. Studies on animal models do not always translate in the clinic; fewer studies on miRNAs have been performed involving human subjects with neuropathic pain, with differing results depending on the specific aetiology underlying neuropathic pain. Further studies using human tissue and liquid samples (serum, plasma, saliva) will help highlight miRNAs that are relevant to neuropathic pain diagnosis or treatment, as biomarkers or potential drug targets.

## 1. Introduction

The perception of pain allows organisms to learn from noxious experiences and avoid harmful stimuli. It relies on the communication between neurons arising from the periphery of the body, which contain receptors specialised in sensing particular nociceptive inputs (chemical, mechanical, thermal), and neurons in the spinal cord, which relay the message to higher brain centres, where the signal can be processed and integrated with other sensory information. Impairment of this mechanism due to lesions to the somatosensory system results in a persistent perception of pain even in absence of painful stimuli. This condition is broadly termed neuropathic pain and includes chronic pain states caused by damage to nerves due to physical trauma and other conditions such as diabetes, cancer, herpes infection and multiple sclerosis. Currently, first-line treatments include tricyclic antidepressants, selective serotonin–noradrenaline reuptake inhibitors and calcium channel modulators (gabapentin and pregabalin); however, the efficacy is limited and varies considerably across patients, while side effects can be substantial [1,2].

Meanwhile, advances in the understanding of a class of regulatory short non-coding RNA molecules, known as microRNA (miRNA), have provided a new hope for treatment of various conditions including cancer, hepatitis C and cardiovascular diseases. Selected miRNA therapeutics are undergoing pre-clinical to phase II clinical trials [3], while the first small interfering RNA (siRNA) therapeutic received FDA approval in 2018 for the treatment of a rare polyneuropathy condition [4]. MicroRNAs are capable of inhibiting the translation of messenger RNA into proteins, inducing widespread changes in the proteome of a cell. In the past decade, increasing evidence of their role in neuropathic pain has surfaced and has helped to identify molecular pathways and mechanisms important in the generation and persistence of chronic pain states [5,6].

The first evidence of a relationship between pain and microRNAs was found by Bai, et al. [7] in 2007, who identified seven dysregulated miRNAs following CFA injection in the masseter muscle of rats. The importance of miRNAs in inflammatory pain was shown in mice by Zhao, et al. [8] through the nociceptor-specific deletion of Dicer, an enzyme responsible for pre-miRNA cleavage into mature miRNA (Figure 1), which caused an attenuated response to CFA intraplanar injection compared to wild type. Several mRNA transcripts were found to be altered in the Dicer null mice, including pain-relevant molecules such as Na_v_1.7, 1.8 and 1.9, and P2X3. Surprisingly, these transcripts were downregulated, suggesting that loss of miRNAs may lead to upregulation of transcriptional repressors which control the expression of these nociceptive genes [8].

Several miRNAs and their respective targets have been investigated in animal models of neuropathic pain during the past decade (Table 1). miRNA targets can be predicted using in silico models; however, experimental validation is necessary to prove the presence of interaction in biological systems and their physiological relevance. A target is deemed to be validated when (a) a reporter assay shows mRNA/miRNA interaction, (b) miRNA/mRNA co-expression is observed with RT-PCR (and possibly in situ hybridisation), (c) miRNA overexpression/knockdown has an effect on the target protein and its biological function [9].

The identified microRNAs and their putative targets belong to a wide variety of pain-relevant pathways that can be broadly classified into those related to neuronal excitability, intracellular signalling and interaction with non-neuronal cells. In this review, examples from each category will be presented. Another important question is how miRNAs are regulated themselves. The study of miRNAs involved in pain has contributed to our understanding of mechanisms underpinning modulation of miRNA levels, such as by sequestration in stress granules (assemblies of untranslated mRNA and RNA binding proteins formed in conditions of stress) and sponging by long non-coding RNA. Additionally, an alternative mode of action of miRNA, targeting proteins directly and causing their activation, will be presented in the context of pain. Finally, opportunities and difficulties relating to the application of miRNA therapeutics to the clinic will be discussed. Examples of miRNAs linked with chronic pain that have been identified in clinical human samples are shown in Table 2.

## 2. MicroRNA Biogenesis

MicroRNAs are transcribed by RNA Polymerase II—and occasionally by RNA Polymerase III—into a hairpin-like structure denominated primary miRNA (pri-miRNA) [96]. Drosha cleavage of the pri-miRNA leads to the formation of pre-miRNA, which is exported out of the nucleus via Ran-GTP and Exportin-5.

In the cytoplasm, Dicer, coupled with the RNA binding protein (TRBP), performs an additional cleavage at the loop terminal, resulting in the separation of a sense and antisense strand, which is commonly degraded. The resulting single strand of miRNA binds to AGO proteins, creating the RISC complex which is also stabilised by several other proteins including FMRP, R2D2 and the germin family.

Finally, the RISC complex binds to the 3′ UTR sequence of the target mRNA, preventing its translation [97]. The consensus sequence, named the “seed” region, is usually positioned between nucleotides 2 to 7 of the 5′ end of the miRNA. Translational inhibition can occur in three ways: mRNA degradation, translational repression and site-specific cleavage [6].

Non canonical pathways have also been identified, including miRtrons, snoRNA and tRNA-derived miRNA, which are generated without Drosha cleavage, and miR-451, which is generated in a Dicer-independent way [98].

## 3. MicroRNA Involvement in Pain Pathways

Understanding the role of miRNAs in nociceptive pathways would contribute to a greater understanding of neuropathic pain development, enabling the identification of new druggable targets. Many studies have been performed in animal models to investigate the role of miRNAs and identify their targets in vivo. In this section, studies mostly from animal models will be discussed in the context of neuronal excitability, synaptic transmission, intracellular signalling and communication with non-neuronal cells. The potential of translating these studies to human disease will be discussed in Section 5, alongside additional findings regarding miRNAs relevant to pain from human tissues.

### 3.1. Neuronal Excitability

The transmission of information from the periphery of the body to the brain relies on the propagation of action potentials throughout the axon of sensory nerves. This is enabled by the presence of different classes of ion channels in the membrane that set the potential of a cell and allow the propagation of electrical impulses. MicroRNAs can alter the expression of ion channels, resulting in altered neuronal excitability and increased pain perception following nerve injury (Figure 2A).

Voltage-gated sodium channels (VGSC) allow the rapid inward entry of sodium ions that enable the generation and transmission of action potentials. They are located at the nodes of Ranvier on the axonal membrane and in DRGs. Na_v_1.7, one of ten members of the VGSC family, was shown to be targeted by miR-30b and miR-182. These miRNAs are downregulated in the DRG following nerve injury, and, if administered intrathecally or injected in the DRG, respectively, they alleviate mechanical hypersensitivity [57,58]. Resveratrol, a natural polyphenol with anti-oxidant and anti-inflammatory properties, was shown to alleviate pain following CCI through miR-182 upregulation, which by inhibiting Nav1.7 may reduce neuronal excitability [56].

Another member of the VGSC family, Na_v_1.3, is targeted by miR-96 and miR-384 in DRGs: when the miRNA mimics are administered intrathecally, the behavioural response indicative of hypersensitivity following nerve injury is alleviated [54,55]. Finally, miR-7a targets the β2 subunit of VGSCs, which is involved in transport and gating of sodium channels and is downregulated in neuronal cell bodies following nerve injury, causing hyper-excitability in C-fibres [99].

Another class of ion channels important in pain transmission are calcium channels, classified in various subtypes, which have different roles in excitable cells. Ca_v_1.2 L-Type calcium channels regulate long-term changes in gene expression, inducing neuronal sensitisation in neuropathic pain. miR-103 modulates three Ca_v_1.2 subunits (Ca_v_1.2-α1, α2δ1 and β1) in spinal cord neurons, promoting mRNA decay. miR-103 is downregulated following nerve injury, concomitant with the upregulation of Ca_v_1.2. Additionally, delivery of miR-103 mimic after SNL reduces pain sensitisation via Ca_v_1.2 downregulation [17].

In addition to sodium and calcium channels, potassium channels also play an important role in neuronal excitability. Voltage-gated potassium channels play a role in setting the resting membrane potential, enabling the neuron to return to its resting state following depolarisation [100]. The miR-17-92 cluster was found to be upregulated following nerve injury and target several voltage-gated potassium channels (Kv1.1, Kv1.4, Kv3.4, Kv4.3, Kv7.5). miR-17-19 AAV injection causes the suppression of A-type potassium currents and induces mechanical allodynia. Administration of potassium channel activators reverses mechanical allodynia following miR-17-92 AAV injection, indicating that the cluster acts through the modulation of potassium channels [37]. More recently, Kv1.2 has been shown to be downregulated by miR-137a in neuropathic pain, contributing to pain hypersensitivity [38]. The authors have shown that cultured nociceptors from rats with CCI to the sciatic nerve display reduced potassium currents and increased neuronal excitability, which was rescued in nociceptors from rats treated with miR-137 antagomir. miR-137 inhibition following CCI resulted in increased Kv1.2 protein expression and pain relief [38].

Another potassium channel, TREK1, is regulated by miR-183. TREK1 is a mechano- and temperature-sensitive channel and contributes to the initiation of action potentials in C-fibres [101]. miR-183 is downregulated following nerve injury, whilst TREK1 is upregulated. miR-183 administration alleviates neuropathic pain through the inhibition of TREK1 [75].

Overall, the evidence suggests that several classes of ion channels are regulated by miRNAs. Due to the multitude of potential targets, miRNA-based therapeutics might also struggle to achieve specificity. However, studying miRNA dysregulation in chronic pain states will help to identify novel pathways that can be targeted pharmacologically with fewer side effects. Modulation of ion channels with specific inhibitors has proved to be a successful strategy in the development of analgesics. For example, pregabalin and gabapentin, currently first-line treatments for neuropathic pain, are calcium channels modulators. However, due to the presence of off-target effects and limited efficacy, there is an unmet need for better inhibitors [102].

### 3.2. Synaptic Transmission

Following the initiation and propagation of action potentials, the signal reaches the central terminals of nociceptors and is relayed to second-order neurons in the spinal cord through synaptic transmission. Modulating the components of this process allows fine-tuning of the amount of information that is transmitted further in the pathway. Various miRNAs have been shown to target genes involved in synaptic transmission in animal models, as shown in Figure 2B. For example, miR-190a is downregulated in diabetic peripheral neuropathy and targets SLC17A6, also known as VGLUT2, a glutamate transporter important in fast synaptic transmission [61]. Similarly, miR-34a undergoes long-term downregulation following CCI and targets VAMP2, belonging to the family of SNARE proteins, important for neurotransmitter release [78]. A study on cancer pain highlighted the role of miR-124 as a regulator of synaptopodin, a structural component of dendrites involved in the formation of strong excitatory synapses [69]. miR-500 is involved in chemotherapy-induced neuropathy, targeting GAD67, an enzyme necessary for the synthesis of GABA in inhibitory interneurons in the dorsal horn of the spinal cord. After paclitaxel injection, miR-500 was upregulated, causing a downregulation of GAD67. This resulted in a loss of inhibitory inputs, leading to an increase in excitatory nociceptive signals [25]. Both miR-124 and miR-500 dysregulation have been linked to nerve injury-induced neuropathic pain [88,103].

Overall, the evidence suggests that following the establishment of neuropathic pain, synaptic transmission is facilitated by the downregulation of miRNA targeting synaptic components, resulting in enhanced pain perception. Another mechanism that has been investigated in neuropathic pain is the loss of inhibitory inputs due to apoptosis of GABAergic interneurons [104]. The involvement of miRNA in this context should be investigated, as the initiation of apoptotic mechanisms requires extensive alteration of gene expression which could potentially be under the control of miRNAs, due to their ability to target multiple transcripts. Indeed, several miRNAs have already been linked to neuronal cell death in neurological diseases such as Alzheimer’s [105], neuroblastoma [106] and ischemic disease [107].

### 3.3. Intracellular Signalling

A fundamental aspect in the establishment of neuropathic pain is the change in gene expression and post-translational modifications in the nociceptors which lead to a persistent alteration in neuronal excitability, a process known as peripheral sensitisation. This occurs through the activation of intracellular signalling pathways, such as the mitogen-activated protein kinases (MAPK), that are triggered by the activation of receptors by inflammatory mediators and cause the activation of kinases and transcription factors, resulting in the upregulation of ion channels and pro-nociceptive receptors (e.g., Na_v_1.8 and TRPV1) [108]. The MAPK family is divided in three pathways: extracellular signal-regulated kinases (ERK), p38 and c-Jun N-terminal kinase (JNK) [109]. Various miRNAs have been found to be involved in the regulation of each of them in the context of pain (summarised in Figure 3).

The ERK pathway is activated in response to inflammation and growth factors, such as NGF, BDNF and other neurotrophins. It involves the sequential activation of Ras, Raf, MEK and ERK and leads to neuronal sensitisation [108]. MicroRNA-144 is downregulated in neuropathic pain and targets RASA1, a member of the Ras family, involved in the ERK activating cascade. Overexpression of miR-144 reverses nerve injury-induced mechanical allodynia and thermal hyperalgesia and leads to a reduction in pro-inflammatory cytokines in the DRG. The interaction between miR-144 and RASA1 was confirmed in satellite glial cells isolated from murine DRG. Furthermore, when miR-144 agomir was administered along with RASA1 lentiviral overexpression, the behavioural response to pain was indicative of increased hypersensitivity and the expression of pro-inflammatory cytokines was restored [52]. Another study highlights RASA1 interaction with miR-206 in PC12 cells and finds that miR-206 is also downregulated following CCI [110]. This study lacks the characterisation of miR-206 effect on RASA1 expression in vivo. However, it emerges that regulation of RASA1 by miRNAs plays an important role in neuropathic pain. miR-206 downregulation is also involved in neuropathic pain through brain-derived neurotrophic factors (BDNF) interaction, leading again to the activation of the MEK/ERK signalling pathway. Overexpression of miR-206 reverses the behavioural response to pain, leading to decreased hypersensitivity, and reduces neuroinflammation, resulting in decreased BDNF and MEK/ERK phosphorylation. When BDNF is also overexpressed, the effects of miR-206 injection are abrogated, concomitant with an upregulation of phospho-MEK and phospho-ERK [111]. Overall, ERK activation in neuropathic pain is facilitated by the downregulation of specific miRNAs, including miR-206 and miR-144.

Another important signalling pathway in the context of neuropathic pain is mediated by p38 activation in spinal microglia, following cellular stress and binding of inflammatory mediators to extracellular receptors [112]. Phosphorylated p38 is translocated into the nucleus where it activates transcription factors, such as NF-kB and STAT3, promoting the expression of inflammatory mediators such as TNFα and IL-1β [108].

Several regulators and downstream targets of p38 are targeted by miRNA. For example, miR-15/16 targets GRK2, a G-protein coupled receptor (GPCR) that inhibits p38. This miRNA cluster is upregulated following CCI in rats and its inhibition leads to reduced hyperalgesia, increased GRK2 expression and decreased phosphorylation of p38 and NF-kB p65. GRK2 knockdown leads to the abrogation of mir-15/16 inhibition, including restoration of p-p38 and p-NF-kB levels [26].

Similarly, miR-221 targets suppressor of cytokine 1 (SOCS1) and its upregulation following CCI causes increased NF-kB and p38 phosphorylation [64]. SOCS1 is also targeted by miR-155, upregulated following CCI [63]. miR-155 inhibition partially reverses changes seen in behavioural responses to thermal and mechanical stimuli, indicating a reduction in hyperalgesia. It also reduces phosphorylation of Ik-Bα and p38, whilst if SOCS1 was also knocked down, the effects of miR-155 inhibition are abrogated. These findings suggest that SOCS1 is involved in neuropathic pain through the regulation of the NF-kB and p38 pathways and is itself targeted by miR-155 and miR-221. miR-155 is also upregulated in neuropathic pain induced by the chemotherapeutic agent Bortezomib, causing increased JNK and p38 phosphorylation [113]. Inhibition of miR-155 causes a reduction in the behavioural response to mechanical stimuli and restoration of JNK and p38 phosphorylation levels [113].

Fewer studies have been performed on the role of JNK in chronic pain. Its activation in sensory neurons and spinal glia following nerve injury is known to be involved in the development of allodynia and hyperalgesia [114]. MicroRNA modulation of JNK, as well as other MAPKs, plays a role in trigeminal nerve injury induced by CCI to the infraorbital nerve. miR-223 is downregulated following CCI, along with an increase in ERK, p38 and JNK phosphorylation levels. miR-223 overexpression with a lentiviral vector improves the behavioural response to mechanical stimuli and restores MAPKs phosphorylation levels. The effect of miR-223 may occur through the direct interaction with the translational repressor MKNK2, involved in the MAPK pathways [42].

Finally, several miRNAs (miR-93, miR-98 and miR-544) have been found to target STAT3 and to be downregulated following nerve injury. STAT3 is an inflammatory transcription factor activated following phosphorylation by p38 [66,67,68]. miR-7a also indirectly regulates STAT3 by targeting neurofilament light polypeptide (NEFL), alleviating neuropathic pain [115].

ERK and p38 are only two of the pathways that are potentially targeted by miRNA to contribute to neuropathic pain. For example, bioinformatic evidence shows that miR-500 potentially targets proteins and transcription factors activated in response to cAMP alterations, another important secondary intracellular messenger [116]. Overall, there is now evidence that several miRNAs influence neuronal excitability through the regulation of intracellular signalling cascades. This could be an interesting route to investigate potential new analgesics. It also provides greater understanding of these mechanisms, which are relevant to a wide variety of human diseases.

### 3.4. Interaction with Non-Neuronal Cell Types

Non-neuronal cell types, including glial cells and immune cells, make up the environment surrounding neurons and contribute to the establishment of inflammation and alteration of excitability following nerve injury [117]. MicroRNAs can influence this environment through various means, inducing changes in several types of non-neuronal cells including macrophages, microglia, Schwann cells, astrocytes and lymphocytes.

#### 3.4.1. Macrophages Communicate through miRNA-Containing Extracellular Vesicles

At the nerve injury site, macrophages are activated by pathogen-associated molecular patterns (PAMP) and damage-associated molecular patterns (DAMP), released in response to tissue injury. They contribute to the increased recruitment of immune cells via the secretion of MMPs and vasoactive mediators and establish tissue inflammation through the secretion of cytokines, interferons, interleukins, prostanoids, growth factors and nitrous oxide. Macrophages are also recruited in the dorsal root ganglia following nerve injury and cause additional secretion of pro-inflammatory cytokines that ultimately lead to a change in post-translational modifications and gene expression in neuronal cell bodies, resulting in increased excitability [118].

In addition to the secretion of cytokines and other well-known pro-inflammatory mediators, macrophages communicate with neuronal cell bodies through miRNA-containing exosomes. McDonald [83] showed that exosomes secreted by activated murine macrophages display a characteristic miRNA signature and induce NF-kB activation, a known pro-inflammatory intracellular mediator, in vitro. In vivo studies show that intraplanar injection of exosomes from LPS-treated macrophages induces increased thermal hyperalgesia shortly after CFA administration. However, over the course of 48 h, exosomes from both LPS+ and LPS- macrophages caused a decrease in the severity of thermal hyperalgesia. This indicates that macrophage activation might promote transient increase in inflammation and hyperalgesia, followed by resolution. Additionally, human patients suffering from CRPS show an increased level of miRNA in circulating exosomes, attesting the potential importance of these molecules.

Another piece of evidence supporting the concept of communication between neurons and macrophages through miRNA-containing exosomes following nerve injury has been described by Simeoli, et al. [119]. Cultured DRG neurons treated with capsaicin release exosomes which contain miR-21. When these exosomes are administered to macrophages, a switch to M1 pro-inflammatory phenotype is observed which is mediated by miR-21. In vivo miR-21 antagomir intrathecal injection causes decreased hyperalgesia and a lower number of M1 macrophages. This indicates that miR-21, secreted in exosomes by activated neuronal cell bodies, stimulates a pro-inflammatory phenotype in macrophages, which leads to increased inflammation and hyperalgesia. Zhang, et al. [13] identified a similar mechanism involving miR-23 and A20, a negative regulator of NF-kB which can induce M1 polarization. miR-23 was shown to be secreted in extracellular vesicles (EVs) from cultured DRG neurons explanted from SNI mice or following capsaicin administration. Macrophages cultured with miR-23-containing EVs displayed a downregulation of A20 protein level, increased NF-kB activation and increased expression of the M1 marker Nos2. Inhibition of miR-23 following SNI in mice resulted in reduced hyperalgesia and inflammation, as well as reduced macrophage infiltration in the DRG, which were more likely to display an M2 non-inflammatory phenotype.

#### 3.4.2. Schwann Cell Proliferation Is Directed by miRNA

Schwann cells (SCs) are found in the peripheral nervous system wrapped around neuronal axons. Their role involves myelinating large-diameter axons and ensheathing small-diameter axons in structures known as Remak bundles, providing nourishment to the neurons and promoting peripheral nerve regeneration [120]. Following nerve injury, the interaction between Neuregulin on the axonal membrane and ERBB2/3 on SCs leads to demyelination, followed by SC proliferation. Additionally, SCs secrete NGF, BDNF, prostaglandins, MMPs and cytokines, which lead to peripheral sensitisation and altered neuronal gene expression [121].

The SC response following nerve injury consists of a de-differentiation and proliferation phase, followed by a remyelination phase. Several miRNAs have been implicated in the proliferation phase as described in a review by Sohn and Park [122]. Examples include let-7b, implicated in NGF inhibition and downregulated following nerve injury, [123] and miR-1, targeting BDNF [124].

A study by Viader [125] analysed miRNA expression in the distal stump of injured nerves in mice, largely constituted by SCs, and identified 48 dysregulated miRNAs. Bioinformatic analysis showed that 31 of the dysregulated miRNAs’ predicted targets are involved in SC de-differentiation and proliferation. Additionally, deletion of Dicer resulted in a delay in the proliferation and remyelination cycle. A closer look at the identified miRNAs revealed that miR-34 is downregulated sharply following nerve injury and targets Notch1 and Ccnd1, positive regulators of SC proliferation. Interestingly, miR-140, which is downregulated until day 7 post-injury, was shown to target Egr2, a transcription factor directing myelination, in in silico and cell culture systems; however, their expression in vivo is positively correlated. This could point to miR-140’s role in fine-tuning Erb2 expression, or it could indicate that in vivo interaction does not actually occur to a significant extent.

Several studies have investigated the role of miRNA expressed in SCs in nerve regeneration following injury; however, studies focused on miRNA in SCs that contribute to the establishment or persistence of neuropathic pain are lacking. Given the role that SCs play in nerve regeneration, it is difficult to distinguish between adaptive changes that promote regeneration and maladaptive ones that lead to chronic pain. The comparison between neuropathic models characterised by different pain intensities, such as the one presented by Norcini, et al. [126], or the comparison between genetic lines with different susceptibilities to pain, as shown in Bali, et al. [127], could be used to overcome this hurdle.

#### 3.4.3. Lymphocyte Differentiation Is Altered by miRNA Expression

Lymphocytes, in particular, T cells, are recruited to the injury site following nerve damage where they secrete pro-inflammatory cytokines. They also infiltrate the spinal cord and promote astrocytic activation [118].

A study using blood samples collected from patients suffering from a variety of neuropathic pain conditions (polyneuropathy, postherpetic neuralgia and trigeminal neuralgia) identified two differentially expressed miRNAs compared to healthy individuals: miR-124a and -155. They are both involved in the differentiation of regulatory T cells (Tregs). miR-124a and -155 directly repress histone deacetylase sirtuin1 (SIRT1), which is a negative regulator of Foxp3, a master regulator of the development of Tregs. Increased levels of miR-124a and miR-155 in neuropathic pain promote Tregs differentiation, which have an anti-inflammatory role [90].

In a mouse model for type II diabetes-induced peripheral neuropathy, miR-590-3p was found to be downregulated in the dorsal root ganglia, alongside an increase in CD4 immunolabelling compared to non-diabetic mice, indicating increased T cell infiltration. Overexpression of miR-590 attenuated the behavioural responses indicative of pain. The validated target RAP1A was shown to promote T cell viability and migration in isolated T cells, indicating that neuropathic pain in diabetic mice might be exacerbated by RAP1A-induced T cell infiltration, enabled by the downregulation of miR-590-3p [51].

#### 3.4.4. Microglial Activation Is Regulated by miRNA

Microglia are essentially CNS-specific macrophages and play an important role in inflammation and nociception in the spinal cord. Following nerve injury, they show increased activation in the spinal cord, characterised by increased phosphorylation of p38 and ERK1/2, increased expression of M1 markers (MHC Class II, CD45, integrins) and a morphological change [118].

MicroRNA-124 has been identified as an important regulator of the transition to the activated pro-inflammatory phenotype of microglia. Its expression is highest in non-activated microglia and its knockdown results in in vivo microglial activation [128]. This plays a role in the transition from acute to persistent hyperalgesia: downregulation of miR-124 in isolated spinal cord microglia following plantar IL-1β injection is observed concomitantly with an increase in M1 markers. Additionally, the injection of miR-124 reverses mechanical allodynia in mice following spared nerve injury [103]. Similarly, miR-451, which is downregulated during inflammatory pain, prevents microglial activation by targeting TLR4, a receptor on microglial surface responsible for cytokine release [72].

Regulation of microglial activity by miRNA in neuropathic pain has been also shown to occur through autophagy impairment. Increased miR-195 in spinal microglia is correlated with a decrease in autophagy activation following peripheral nerve injury. In particular, miR-195 targets ATG14, involved in the autophagy pathway. Impairment of autophagy leads to further accumulation of pro-inflammatory cytokines and exacerbation of hyperalgesia [12].

#### 3.4.5. Astrocytic Activation and Cytokine Production Is Influenced by miRNA

Astrocyte activation in the spinal cord is observed with some delay compared to the microglial response, but its increase is sustained for up to 5 months following nerve injury. A role for miRNA-mediated communication between glia and neurons has been reported [18], in which miR-186, CXCL13 and its receptor CXCR5 trigger astrocytic activation. Following spinal nerve ligation (SNL), CXCL13 expression is increased in neurites, whilst miR-186 is downregulated. miR-186 inhibition in naïve mice elicits behavioural responses to a lower threshold of mechanical stimulation, which is reversed by CXCL13 neutralising antibody injection. Additionally, CXCR5, a known CXCL13 receptor expressed in astrocytes, is upregulated following SNL and its inhibition leads to reduced SNL-induced pain hypersensitivity for up to 91 days. In CXCR5 KO mice, decreased glial activation is observed. This indicates that CXCL13 expression in neurons is controlled by miR-186 and leads to CXCR5 activation in astrocytes, which triggers glial activation through the ERK signalling pathway.

Once astrocytes are activated, miR-146 plays a role in establishing the balance in pro-inflammatory molecule production. Inflammatory mediators such as TNFα and IL-1β, which stimulate astrocytic activation, lead to a sharp increase in TRAF6 followed by a delayed miR-146 increase. miR-146 targets TRAF6 mRNA, inhibiting its downstream targets JNK and CCL2. Simultaneously, LPS treatment enhances AP-1 (downstream of JNK) binding site for miR-146, promoting its transcription. It emerges that miR-146 is at the centre of a negative feedback loop in astrocytes to fine-tune the production of pro-inflammatory signalling molecules such as JNK and CCL2 [129].

## 4. A Closer Look at miRNA Regulation and Function

In this review, several examples of how miRNA dysregulation contributes to the development of neuropathic pain are summarised. Typically, this involves the downregulation of miRNAs that target molecules which promote excitability and inflammation. But how is miRNA expression altered following nerve damage?

A study by Leung [130] found that in conditions of cellular stress, which lead to the formation of stress granules, miRNA-induced repression of translation is decreased five-fold. Additionally, Ago2, part of the RISC complex, is co-immunoprecipitated with PARP-13, a component of stress granules. This indicates that following stress, Ago2 bound to miRNA might be sequestered to stress granules and prevented from repressing its mRNA targets. Similar evidence was found in the context of neuropathic pain. Aldrich, et al. [131] identified a miRNA cluster–miR-96/182/183–which is downregulated following nerve injury and localised at the periphery of cell bodies. In contrast, miR-96/182/183 displayed an even localisation throughout the cell bodies of uninjured animals. In injured neurons, the miRNA cluster was also co-localised with a stress granule protein (TIA1), which is commonly found in apoptotic neurons. Co-immunoprecipitation and reporter assays are necessary to establish if direct interaction between the miRNA cluster or RISC complexes and TIA1 occurs. However, this provides a potential mechanism through which miRNA regulation takes place during the development of chronic pain, whereby miRNAs are sequestered by stress granules at the periphery of the cell. The author suggests two possible strategies on how this occurs: miRNAs bound to stress granules might become translational activators, or their compartmentalisation at the periphery might result in increased cytoplasmic volume free of translational repressors.

Another way through which miRNAs are sequestered in neuropathic pain is through “sponging” by long non-coding RNA (lncRNA). This process has also been observed in cancer, as several lncRNAs have been shown to bind to miRNAs in order to derepress their mRNA targets [132]. This is also relevant in the development of neuropathic pain, as reviewed in Song, et al. [133]: examples include miR-381, negatively regulated by lncRNA NEAT1 [31], miR-150 and miR-154 by XIST [73], miR-206 and miR-129 by MALAT1 [27,81], and miR-124 and miR-141 by SNHG16 [36]. A study by Wang, et al. [134] investigated the network of interactions between lncRNAs, miRNAs and mRNAs in Schwann cells cultured from rats affected by diabetic peripheral neuropathy. The expression of thousands of transcripts was analysed with RNA sequencing. The resulting competing endogenous RNA (ceRNA) network was calculated using co-expression similarity and target prediction algorithms and included 38 lncRNAs, 10 miRNAs and 702 mRNAs. This complex network of interaction should be validated experimentally; however, the interaction between one of the miRNAs involved (miR-212-5p) and its predicted target Gucy1a3 was verified in a cell-line. ceRNA networks were also investigated in trigeminal neuropathic pain in mice by Fang, et al. [135]. Total RNA sequencing identified 67 miRNAs, 216 lncRNAs, 14 circRNA and 595 mRNA that were dysregulated following infraorbital CCI compared to sham-operated mice. Computational analysis identified a complex network of interactions, involving several previously identified pain-related transcripts. It is perhaps unlikely that the network of interactions will be exactly replicated in humans; however, studies of this kind are invaluable to identify master regulators that dictate molecular changes leading to neuronal hyperexcitability in chronic neuropathic pain, helping to identify a set of targets that can then be investigated in humans.

The complexity surrounding miRNA regulation does not end here. The conventional modality of action of miRNA consists in translational repression of mRNA targets. However, an alternative mechanism of action has emerged, which involves the direct interaction between miRNA and a protein, resulting in its activation. Lehmann, et al. [105] found that let-7b can be secreted extracellularly by dying neurons and induce further neuronal cell death through TLR7 activation. In the context of pain, let-7b induces nociception through direct interaction with TLR7, which elicits TRPA1-mediated currents. Let-7b has been observed to be co-localised on the cell membrane with TRPA1 and TLR7, but not in TLR7 deficient cells, indicating that let-7b directly binds to TLR7 and promotes TRPA1 activation [136]. In other words, let-7b has a positive effect on TRPA1 through direct interaction with TLR7′s amino acid sequence. Mir-21 and miR-29a also bind to human TLR8 and murine TLR7 in non-small cell lung cancer, increasing the secretion of pro-inflammatory cytokines [137]. Indeed, miR-21 is upregulated in mouse following SNL and targets TLR8, causing ERK activation [74].

Another miRNA acting in a similar way is miR-711, which directly interacts with the extracellular loop of TRPA1 to elicit pruritus. miR-711 cheek injection is sufficient to cause pruritus in mice, and this requires the presence of TRPA1 and the core miR-711 sequence (GGGACCC). Calcium imaging of cultured DRG neurons show that extracellular miR-711 administration is sufficient to elicit currents in a subset of neurons, whilst the presence of a TRPA1 inhibitor prevents this. Again, this indicates that miR-711 directly interacts with TRPA1 peptide sequence causing its activation [138].

This mode of miRNA interaction presents an additional opportunity to harness with pathways involved in chronic pain and inflammation that could be exploited in drug discovery. It also provides a fascinating insight on the regulatory capabilities of noncoding RNA molecules. Further research is needed to identify which other miRNAs (or other non-coding RNA), if any, act in a similar way and how the interaction causes receptor activation.

## 5. Potential Clinical Application of miRNA Studies

MicroRNA expression appears to be dynamic, changing extensively according to the pain model, time frame and animal studied. In a systematic study by Guo, et al. [139], data from 37 different papers on miRNA expression in rat neuropathic pain models (SCI, CCI and SNL) were compared. Despite the use of similar pain models, few studies showed overlapping expression patterns. Only five dysregulated miRNAs (rno-miR-183/96, rno-miR-30b, rno-miR-150 and rno-miR-206) were observed in two or more studies, suggesting that various factors including variations in the animal weight, age, time after injury, injury model and handling could influence significantly miRNA expression. This entails a significant obstacle in the application of miRNA research to the clinic. However, it could contribute to the development of strategies to stratify patients, as particular miRNA signatures could be associated with a better response to particular drugs and treatments or with chronic pain of different origins.

An example of this is presented by Dayer, et al. [89], who performed a study on circulating miRNAs in plasma of patients affected by chronic pain following orthopaedic trauma. The pain symptoms were classified into neuropathic, nociceptive, mixed and CPRS through a DN4 questionnaire, clinical history investigation and assessment by physiotherapists and physicians. Two circulating plasma miRNAs (miR-320 and miR-98) were found to successfully classify patients suffering from neuropathic or nociceptive pain in 70% of the cases. If more miRNAs are identified, possibly through wider RNA-sequencing screening, the accuracy might increase, leading to faster diagnosis, earlier start of appropriate treatment and a more robust and efficient cohort selection in clinical trials.

The role of circulating miR-320 in chronic post-traumatic pain in humans has also been elucidated by Linnstaedt, et al. [140], pointing to an alternative approach to stratify patients based on genetic variants. A riboSNitch (i.e., a regulatory element in an RNA transcript with a function that is disrupted by a specific SNP) in the 3′UTR of the FKBP5 gene was linked to increased vulnerability to chronic post-traumatic pain by decreasing the affinity of miR-320 to its binding region. Individuals with the minor allele were more likely to report symptoms of musculoskeletal pain 6 weeks post-trauma if they experienced high levels of peritraumatic distress, while the level of distress in individuals with the major allele had little impact on the development of chronic pain. This riboSNitch was found to affect the secondary structure of FKBP5 3′UTR, causing the miR-320 binding site to become inaccessible, leading to increased FKBP5 translation, glucocorticoid resistance and susceptibility to chronic pain. This evidence shows how a deeper understanding of the regulatory networks involving miRNA–mRNA interactions could lead to new strategies to identify patients who are at higher risk of developing chronic pain.

Another fundamental question when translating animal studies to humans is whether miRNA function is conserved across different organisms. Studies on human DRG and spinal cord are lacking due to the inaccessibility of the tissues, which makes it challenging to translate the role of miRNAs characterised in animal studies to human disease. Some studies have been performed on blood samples in CRPS patients [83,84] and a variety of neuropathic pain conditions [90], which have been discussed in previous paragraphs.

Interesting translational insights on miRNAs and neuropathic pain come from Leinders, et al. [91]. MicroRNA expression was analysed in WBCs and sural nerve biopsies of patients affected by a variety of peripheral neuropathies. It was found that miR-132-3p was significantly increased in WBCs of neuropathic pain patients compared to healthy controls, and that miR-132-3p expression in the sural nerve was correlated with pain intensity in patients with peripheral neuropathies. Experiments on animal models identified the same miRNA to be altered in the spinal cord of rats 10 days following SNI. Additionally, daily injections of miR-132-3p antagonist caused an increase in mechanical withdrawal thresholds following SNI, indicative of reduced allodynia, compared to rats treated with scrambled control injections. This work highlighted the importance of this particular miRNA, which is altered in both nerve and WBCs of human patients and was shown to affect the behavioural responses to pain in a pre-clinical animal model. Interestingly, miR-132 was also found to be upregulated in plasma of patients affected by peripheral neuropathies in a different study [92]. This type of study is invaluable to understand the local and systemic impact of the development of peripheral neuropathies on gene expression, while the comparison with pre-clinical models allows a more in-depth analysis of the molecular mechanisms that drive the transition to chronic pain.

Similarly, a study by Tavares-Ferreira, et al. [88] used lingual neuroma tissues from patients with lingual nerve injury and pre-clinical models to identify changes in miRNA expression following nerve injury. Lingual nerve injury can occur during routine dental practices such as anaesthetic injections and third molar removal. Following nerve damage, the injured axon undergoes extensive sprouting in the attempt to regenerate, supported by proliferating Schwann cells. However, this process may lead to the formation of a swollen mass—classified as a neuroma—constituted by the accumulation of fibroblasts, immune cells, Schwann cells and collagen fibres. The patient is often left with altered sensation and hyperalgesia, with symptoms including anaesthesia, discomfort, tingling, burning or sharp shooting pain. Repairing the nerve, which involves dissecting the neuroma out and micro-surgically re-joining the two nerve ends, promotes functional recovery in most patients [141]. Investigation of these neuromas revealed two miRNAs (hsa-miR-29a and hsa-miR-500a) that were differentially expressed in painful and non-painful human lingual nerve neuromas, and significant correlations between miRNA expression and the pain VAS score for both miR-29 and miR-500. This comparison allows the identification of factors which are associated with pain intensity, rather than positive adaptive changes, such as factors promoting nerve regeneration.

The study also investigated miRNA expression when a similar type of injury was replicated in rat, which will enable further investigation of the molecular basis of miRNA function. Chronic constriction injury was performed on the rat lingual nerve: two miRNAs were identified (rno-miR-667 and rno-miR-138) that were correlated with altered behaviour indicative of allodynia. Bioinformatics analysis showed several common potential targets, involved in inflammation, chemotaxis and ion channel activity. Despite the difference in the miRNAs identified, the shared potential targets hint that miRNA investigation is useful to identify important pathways involved in the establishment of chronic neuropathic pain.

## 6. Conclusions

Neuropathic pain results from various pathophysiological processes following nerve injury, including inflammation, neuronal death, altered excitability and long-term synaptic alterations, which take place from the periphery of nociceptors to central circuits involved in pain processing. Being such a complex and subjective phenomenon, the mechanisms underlying its development are not fully understood.

The study of microRNAs provides new insights into pathways involved in the development and persistence of neuropathic pain. Several miRNAs have been shown to induce analgesia or hypersensitivity in animal models, through targeting various pathways linked with neuronal excitability, intracellular signalling and communication with glial and immune cells. These molecules can exert their effect in various ways, with the most conventional being through mRNA repression, whilst direct protein activation has also been observed. The study of mechanisms that cause miRNA depletion during neuropathic pain, such as long non-coding RNA sponging and stress granule sequestration, has provided valuable insights on how translational homeostasis is achieved in health and disrupted in disease.

As the current treatments available are not satisfactory for long-term management of neuropathic pain, new targets are needed in order to develop better therapeutics. MicroRNAs could potentially be targeted to achieve analgesia, being theoretically capable of influencing several molecules involved in various pathways. While several miRNA-based therapeutics are in phase I/II clinical trials, none have reached phase III clinical trials yet. Challenges include RNA degradation, low tissue permeability and low specificity [142]. Chemical modifications can increase RNA stability, while issues concerning tissue permeability and specificity can be addressed by designing drug delivery systems that target specific cell types [143].

MicroRNAs also offer the opportunity to investigate phenotypic subtyping, in order to stratify patients into groups that are more or less likely to respond to a certain treatment by their miRNA signature. Finally, the study of miRNAs is critical to develop further understanding of the pathophysiology of neuropathic pain and may benefit several other conditions that share common pathways such as inflammation, neuronal cell death and alterations in intracellular signalling.

Overall, a stronger focus on the use of human tissue in the field of miRNAs and neuropathic pain is needed in order to identify clinically relevant biomarkers and drug targets. With the advent of new technologies and the reduction in sequencing costs, investigation of patients’ samples is increasingly more accessible and represents the most promising route to advance the field and benefit patients, while animal models remain a great tool to discover the mechanisms underlying miRNAs’ action in the context of pain.

## Figures and Tables

**Figure 1 biomedicines-11-00775-f001:**
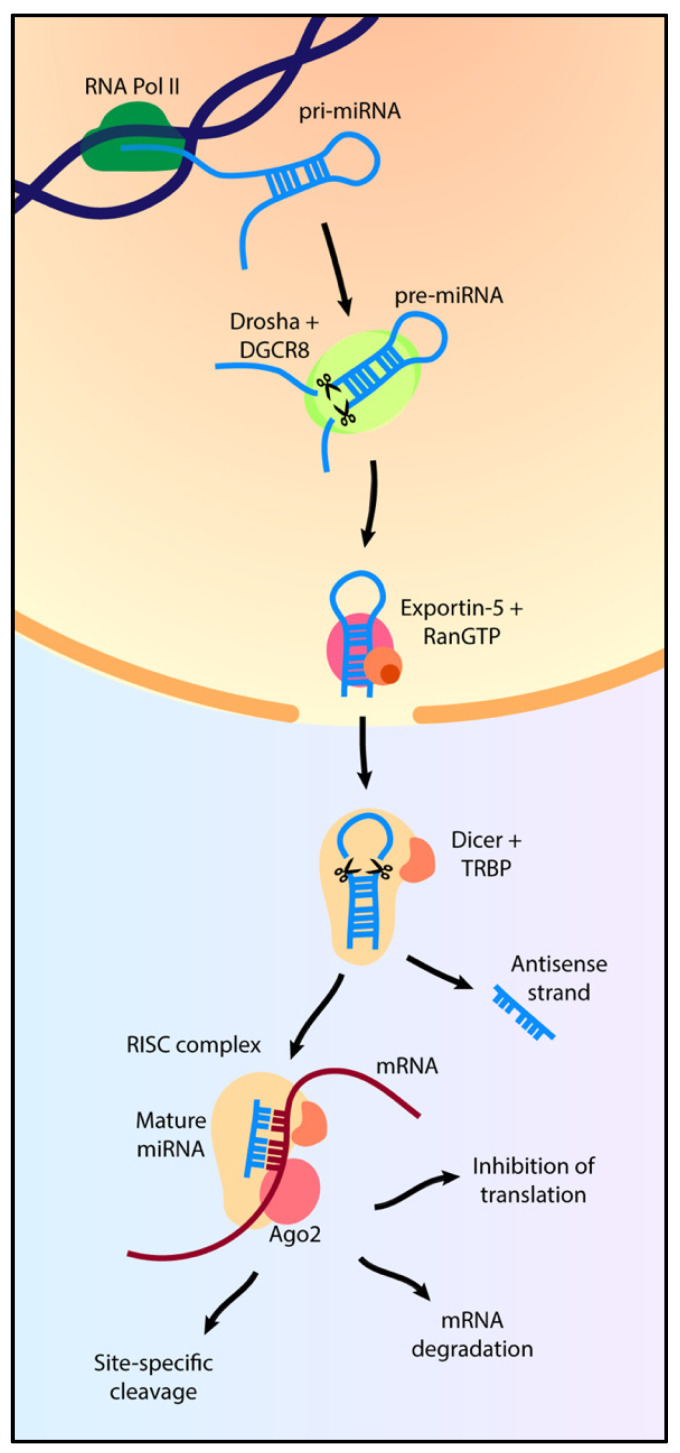
The canonical pathway of miRNA biogenesis.

**Figure 2 biomedicines-11-00775-f002:**
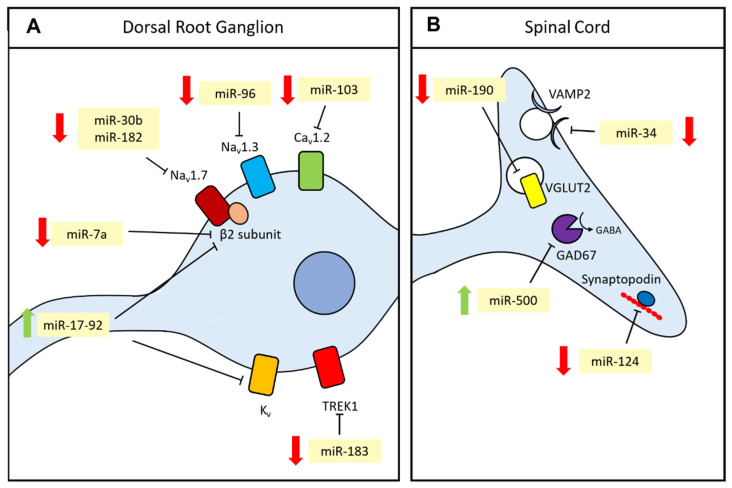
Diagram showing miRNAs dysregulated in neuropathic pain whose targets are directly involved in (**A**) cellular excitability and (**B**) synaptic transmission, verified in animal models. Red downward arrows indicate that the miRNA is downregulated during neuropathic pain, whilst upward green arrows indicate miRNA upregulation in neuropathic pain conditions.

**Figure 3 biomedicines-11-00775-f003:**
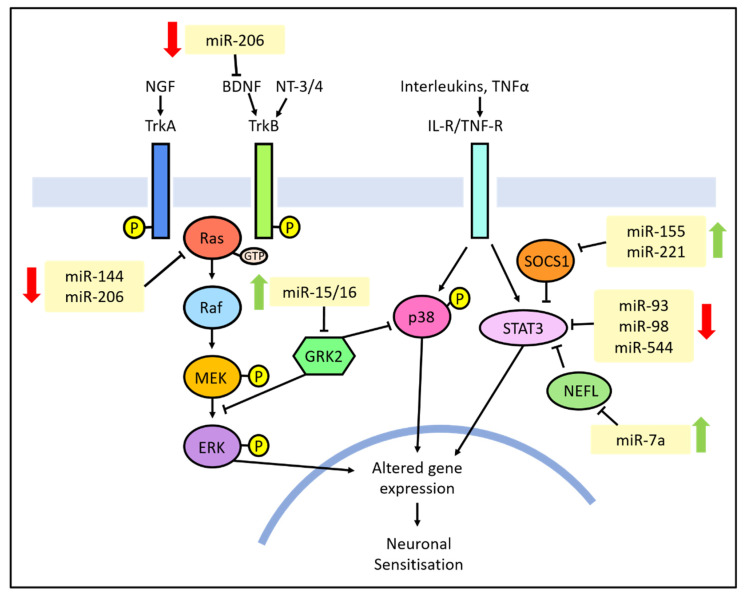
Diagram displaying miRNA dysregulated in neuropathic pain and their targets involved in intracellular signalling. Red downward arrows indicate that the miRNA is downregulated during neuropathic pain, whilst upward green arrows indicate miRNA upregulation in neuropathic pain.

**Table 1 biomedicines-11-00775-t001:** List of dysregulated miRNAs in various chronic pain animal models and the relevant validated targets. bCCI = bilateral CCI, BCP = bone cancer pain, CCI = chronic constriction injury, CIP = chemotherapy-induced pain, CIVP = chronic inflammatory visceral pain, DNP = diabetic neuropathic pain, DRG = dorsal root ganglion, iCCI = infraorbital CCI, pSNL = partial spinal nerve ligation, TG = trigeminal ganglion.

MicroRNA	Change	Injury Model	Localisation	Validated Target/s	Reference
miR-150	Down	CCI	DRG	AKT3	Cai, et al. [10]
miR-20b			Spinal cord		You, et al. [11]
miR-195	Up	SNL	Spinal microglia	ATG14	Shi, et al. [12]
miR-23	Up	SNI	DRG	A20	Zhang, et al. [13]
miR-142	Up	CCI	Sciatic nerve	AC9	Li, et al. [14]
miR-34c	Up	BCP	DRG	Cacna1e (Ca_v_2.3)	Gandla, et al. [15]
miR-219	Down	CFA	Spinal cord	CAMKII	Pan, et al. [16]
miR-103	Down	SNL	Spinal cord	Ca_v_1.2	Favereaux, et al. [17]
miR-186-5p	Down	SNL	Spinal cord	CXCL13	Jiang, et al. [18]
miR-23a	Down	pSNL	Spinal cord	CXCR4	Pan, et al. [19]
miR-124	Down	SNL	Spinal cord and DRG	EGR1	Jiang, et al. [20]
miR-30a-5p	Down	CCI	Spinal cord	EP300	Tan, et al. [21]
miR-211	Down	CIVP	Spinal cord	ERK	Sun, et al. [22]
miR-124-3p	Down	CCI	Spinal cord	EZH2	Zhang, et al. [23]
miR-194	Down	CCI	Spinal cord	FOXA1	Zhang, et al. [24]
miR-500	Up	CIP	Spinal cord	GAD67	Huang, et al. [25]
miR-15a/16	Up	CCI	Spinal cord	GRK2	Li, et al. [26]
miR-129	Down	CCI	Spinal cord	HMGB1	Ma, et al. [27]
miR-141			DRG		Zhang, et al. [28]
miR-142-3p		SNL	DRG		Zhang, et al. [29]
miR-193		DNP	Spinal cord		Wu, et al. [30]
miR-381		CCI	DRG		Xia, et al. [31]
miR-381	Down	CCI	Spinal cord	HMGB1, CXCR4	Zhan, et al. [32]
miR-124-3p	Down	CFA	Spinal cord	IL6R	Liu, et al. [33]
miR-146	Up	CCI	DRG, spinal cord	IRAK1, TRAF6	Wang, et al. [34]
miR-9	Up	DNP	Sciatic nerve	ISL1	Sun, et al. [35]
miR-29a					
miR-124	Down	CCI	Spinal cord	JAG1	Li, et al. [36]
miR-141					
miR-17-92 cluster	Up	SNL	DRG	KCNA1, KCNA4, KCNC4, KCND3, KCNQ5, DPP10, SCN1B	Sakai, et al. [37]
miR-137a	Up	CCI	DRG, spinal cord	KCNA2	Zhang, et al. [38]
miR-216	Down	CCI	DRG, spinal cord	KDM3A	Wang and Li [39]
miR-152	Down	PNI	Spinal cord	MafB	Tozaki-Saitoh, et al. [40]
miR-26a	Down	CCI	Spinal cord	MAPK6	Zhang, et al. [41]
miR-223	Down	iCCI	TG	MKNK2	Huang, et al. [42]
miR-101	Up	CCI	Spinal cord and microglia	MKP4	Qiu, et al. [43]
miR-183	Down	CCI	Spinal cord	mTOR	Xie, et al. [44]
miR-125a-3p	Down	CFA	TG	p38 MAPK	Dong, et al. [45]
miR-195	Up	iCCI	Caudal medulla and CSF	Patched1	Wang, et al. [46]
miR-122	Down	CCI	Spinal cord	PDK4	Wan, et al. [47]
miR-1224	Down	CFA	Spinal cord	pre-circ-Filip1I	Pan, et al. [48]
miR-16	Up	CCI	Spinal cord	RAB23	Chen, et al. [49]
miR-202	Down	bCCI	Spinal cord	RAP1A	Fang, et al. [50]
miR-590		DNP	DRG		Wu, et al. [51]
miR-144	Down	CCI	DRG	RASA1	Zhang, et al. [52]
miR-140	Down	CCI	DRG	S1PR1	Li, et al. [53]
miR-96	Down	CCI	DRG	SCN3A	Chen, et al. [54]
miR-384					Ye, et al. [55]
miR-182	Down	CCI	DRG	SCN9A	Jia, et al. [56]
		SNI			Cai, et al. [57]
miR-30b					Shao, et al. [58]
miR-34	Up	CFA	Spinal cord	SIRT1	Chen, et al. [59]
miR-448		CCI			Chu, et al. [60]
miR-190a	Down	DNP	Spinal cord	SLC17A6, (VGLUT2)	Yang, et al. [61]
miR-135a	Up	CCI	Spinal cord	SLC24A2	Zhou, et al. [62]
miR-155	Up	CCI	Spinal cord	SOCS1	Tan, et al. [63]
miR-221			Spinal cord and microglia		Xia, et al. [64]
miR-218	Up	CCI	Spinal cord and microglia	SOCS3	Li and Zhao [65]
miR-93	Down	CCI	Spinal cord	STAT3	Yan, et al. [66]
miR-98					Zhong, et al. [67]
miR-544					Jin, et al. [68]
miR-124a	Down	BCP	Spinal cord	Synpo	Elramah, et al. [69]
miR-28	Down	CCI	Spinal cord	TF Zeb1	Bao, et al. [70]
miR-30c	Up	SNI	Spinal cord	TGFβ	Tramullas, et al. [71]
miR-451	Down	CFA	Spinal cord and microglia	TLR4	Sun and Zhang [72]
miR-154	Down	CCI	Spinal cord	TLR5	Wei, et al. [73]
miR-21	Up	SNL	DRG	TLR8	Zhang, et al. [74]
miR-183	Down	CCI	DRG	TREK1	Shi, et al. [75]
miR-183	Down	CCI	Spinal cord	TXNIP	Miao, et al. [76]
miR-134	Down	CCI	Spinal cord	Twist1	Ji, et al. [77]
miR-34a	Down	CCI	DRG	VAMP2	Brandenburger, et al. [78]
miR-128	Down	CCI	Spinal cord and microglia	ZEB1	Zhang, et al. [79]
miR-150	Down	CCI	Spinal cord and microglia	Zeb1	Yan, et al. [80]
miR-206	Down	CCI	Spinal microglia	ZEB2	Chen, et al. [81]

**Table 2 biomedicines-11-00775-t002:** List of studies on miRNA linked with chronic pain with a neuropathic component conducted on human samples. CRPS = complex regional pain syndrome, CiPN = chemotherapy-induced painful peripheral neuropathy, DN = diabetic neuropathy, PPN = painful peripheral neuropathy, sDSP = symptomatic distal sensory polyneuropathy, WBC = white blood cells.

Clinical Condition	Control	Sample Types	Significance	Reference
CiPN in patients with multiple myeloma	Myeloma patients without CiPN	Plasma	miR-22, -23a and -24a are clinically relevant biomarkers for CiPN.	Łuczkowska, et al. [82]
CRPS	Healthy	Whole blood	Exosomal miRNA signature is altered in CPRS patients.	McDonald, et al. [83] Orlova, et al. [84]
Diabetic neuropathy	Diabetic patients without neuropathy	WBCs	miR-128 is upregulated in diabetic neuropathy; miR-155 and -499 are downregulated.	Ciccacci, et al. [85]
Diabetic neuropathy	Healthy	Plasma and skin biopsy	miR-199-3p is increased in patients with DN and targets SERPINE2.	Li, et al. [86]
HIV-associated sDSP	Non-DSP and HIV	Plasma	Increased miR-455 expression is associated with reduced neurite growth.	Asahchop, et al. [87]
Lingual nerve injury	Lingual nerve injury without pain	Neuroma	miR-29a and miR-500a are inversely correlated with clinical VAS scores.	Tavares-Ferreira, et al. [88]
Musculoskeletal pain	Healthy	Plasma	miR-320 and miR-98 successfully distinguish the origin of chronic pain in 70% of the patients.	Dayer, et al. [89]
Neuropathic pain of various origins	Healthy	Primary human CD4^+^ T cells	Increased miR-124a and miR-155 levels promote Tregs differentiation.	Heyn, et al. [90]
Painful peripheral neuropathy	Painless peripheral neuropathy and healthy	WBCs and sural nerve biopsy	miR-132-3p is overexpressed in WBC and sural nerve of patients affected by peripheral neuropathies.	Leinders, et al. [91]
Painful peripheral neuropathy	Painless peripheral neuropathy and healthy	Plasma and sural nerve biopsy	miR-101 and -132 are altered in plasma and sural nerve of DN patients. miR-101 targets KPNB1.	Liu, et al. [92]
Painful peripheral neuropathy	Painless peripheral neuropathy and healthy	WBCs, sural nerve and skin biopsy	miR-21, -146 and -155 are differentially expressed in WBC, skin and sural nerve of patients affected by peripheral neuropathies.	Leinders, et al. [93]
Trigeminal neuralgia	Healthy	Serum	miR-132, -146, -155 and -384 are upregulated in patients with trigeminal neuralgia.	Li, et al. [94]
Neuropathic low back pain	Patients without neuropathic pain	Sinuvertebral nerve biopsy	TRPV1 upregulation is inversely correlated with miR-375 and -455 expression.	Li, et al. [95]

## Data Availability

Not applicable.

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
