# Peer review of "The Role of miRNAs in Neuropathic Pain"

_biomedicines, 2023, doi:10.3390/biomedicines11030775_

Round 1

Reviewer 1 Report

Ref: biomedicines-2189989

Title: The role of miRNAs in neuropathic pain

Recommendation: Accept for publication

Very well written review. Nicely described and informative schemes are an additional value. The topic of the review is novel and will certainly find many readers. The literature cited by the authors is current and published mostly after 2017. Therefore, I recommend this review to be publish as it is.

Author Response

We thank reviewer 1 for their kind comments, we are pleased they enjoyed reading our review.

Reviewer 2 Report

I enjoyed reading this manuscript. However, I have few suggestions/clarifications that could improve the manuscript.

Please include critical discussion about reliability of findings from different studies concerning experimental design and technical aspects. For each miRNA discussed in the manuscript, it would be good to know which form of the miRNA was measured to make the findings (pri-, pre-, mature, -5p, -3p) and perhaps also how the measurements were performed. It would also be good to clarify whether the observed effects of the miRNAs on the targets were direct or indirect, if known, and how specific the effects of the miRNAs were, and how tissue specific the miRNAs discussed in the review are, if known. It would also be interesting to include discussion whether any genetic variants are known, which may influence the miRNA or miRNA-target interaction and what are the associated phenotypes or disorders. On 187 - 188 the authors appropriately state that “Due to the multitude of potential targets, miRNA-based therapeutics might also struggle to achieve specifity.” However, in the rows 613 – 614 the authors conclude that "MicroRNAs could potentially be targeted to achieve analgesia, being theoretically capable of influencing several molecules involved in various pathways”. To support this conclusion, it would be good to include discussion how the issues or risks associated with the potential lack of target and tissue specificity could be mitigated.

Table 1 - It is surprising that the authors have chosen to sort miRNAs according to their target. As rightfully pointed by the authors in the text and as evident from the table, miRNAs usually have multiple targets. Sorting miRNAs by their number or by the pain model seems to be more logical. This would also simplify finding a particular miRNA in the table while reading about it in the text. 

Page 6 - when describing miRNA biogenesis, for the sake of completeness and considering the Figure 1 title, I suggest also mentioning that at least some miRNAs can be produced by non-canonical (Dicer-independent) biogenesis pathway. 

Line 150 - the sentence makes a misleading impression that Sirt1 is the only target of resveratrol, while it is known that this compound has rather complex pharmacology and is known as PAINS compound i.e. pan interference assay compound. 

Line 223 - it is worth mentioning that also the role of miRNAs in neurodegenerative diseases, such as PD, have been actively studied, with a few references to such studies. 

Section 3.3 - considering the role of opioids and other GPCR ligands in pain, it is surprising that the authors do not discuss miRNA regulation of GPCR signaling pathways. 

Section 4 - it is worth mentioning that miRNA biogenesis by itself can be compromised in stress conditions (for example, via down-regulation of Dicer levels and/or activity), as has been demonstrated in several studies. 

Line 562 – apparently, “Src” should read as “Scr” or “scrambled”. 

Section 5 – it is worth mentioning that another issue that potentially complicates translation of the animal model findings is a conservation of their target sequences. While the sequences of miRNAs may be conserved, the sequences of their target 3’UTRs may differ. There are at least a few examples in the literature describing human-specific miRNA regulation of some targets, especially in the brain, which may also be relevant to neuropathic pain. 

PubMed search for microRNAs and neuropathic pain finds 95 articles published in 2022. Surprisingly, there is only one reference to a publication from 2022 mentioned in the manuscript.  The text could be updated to include references to new studies.

Author Response

We thank reviewer 2 for the helpful suggestions. We have modified the manuscript and included extra references whenever possible. These are our responses to the main comments:

  • Regarding the addition of information on the reliability of the studies and technical aspects, we have included this information whenever it was relevant and when the data was available. Extensive technical details are beyond the scope of this review, which aims to give a broad overview of miRNAs in neuropathic pain.
  • The effect of genetic variants on the miRNA-mRNA interactions is a very interesting topic, but to our knowledge there aren’t any papers where this is investigated in the context of neuropathic pain. However, we have included a paragraph discussing the work described in Linnstaedt et al (2018), where a genetic variant was identified which affects miR-320 and FKBP5 3’ UTR interaction in humans, causing a higher susceptibility to chronic musculoskeletal pain following traumatic injury.
  • A paragraph discussing the state of miRNA-based therapeutics and potential strategies to improve tissue specificity and RNA stability was also included, with reference to two reviews to direct readers looking for more in-depth knowledge.
  • The arrangement of the table of miRNAs linked with neuropathic pain in animal models was not changed, as we believe that the order we have used provides a more meaningful picture of the effects of miRNA regulation on the development of neuropathic pain. It is true that microRNAs have many potential targets; however, relatively few are experimentally validated and they are often found to be targeted by multiple miRNAs. For example, five miRNAs were found to target HMBG1, three target STAT3 and two miRNAs target SCN3A, SCN9A, SOCS1, all from different publications. This potentially represents a bias in the literature, where only genes that are known to have a role in pain or inflammation are chosen to be further validated. However, with our presentation method it is perhaps easier to visualize the experimentally validated knowledge of this complex network of interactions.
  • A paragraph was added mentioning non-canonical pathways for miRNA biogenesis.
  • The mention to Sirt1 was deleted, Resveratrol was described more generally as an anti-inflammatory and anti-oxidant compound.
  • The role of miRNAs on neurodegenerative diseases is an interesting and broad topic, however, we believe that this would be beyond the scope of the paper and therefore we have not included further references to this.
  • GPCRs act through a variety of intracellular pathways, including MAPK pathways, JAK/STAT and cAMP, which are discussed in the section dedicated to intracellular pathways. The direct effect of miRNA on GPCR translation is less investigated to our knowledge, however, we have now specified that the gene targeted by miR-15a/16, GRK2, encodes for a GPCR.
  • “Src” was corrected to scrambled.
  • Three more papers from 2022 were included in the review.

Reviewer 3 Report

This is a very interesting review about the role of miRNA in Neuropathic Pain. The topic is properly described and the paper i well written. Neuropathic Pain often affects adults and elderly (10.3390/diagnostics11040613) and presents a significant impact on quality of life. Especially in older adults it represents a problem and often drugs create more side effects (10.1017/S1041610217001715). So this review is particular relevant. 
I just suggest to consider this in authors' conclusion.

Author Response

We thank reviewer 3 for their kind comments. We have referenced their suggested article on neuropathic pain in the elderly, however the second reference on the use of proton pump inhibitors and their association with depression didn’t seem to be relevant to the topic, therefore we did not include it.